# Concepts of ferrovalley material and anomalous valley Hall effect

Wen-Yi Tong[1], Shi-Jing Gong[1], Xiangang Wan[2,3] & Chun-Gang Duan[1,4]

Valleytronics rooted in the valley degree of freedom is of both theoretical and technological importance as it offers additional opportunities for information storage, as well as electronic, magnetic and optical switches. In analogy to ferroelectric materials with spontaneous charge polarization, or ferromagnetic materials with spontaneous spin polarization, here we introduce a new member of ferroic family, that is, a ferrovalley material with spontaneous valley polarization. Combining a two-band $\mathbf{k} \cdot \mathbf{p}$ model with first-principles calculations, we show that 2H-VSe$_2$ monolayer, where the spin–orbit coupling coexists with the intrinsic exchange interaction of transition-metal $d$ electrons, is such a room-temperature ferrovalley material. We further predict that such system could demonstrate many distinctive properties, for example, chirality-dependent optical band gap and, more interestingly, anomalous valley Hall effect. On account of the latter, functional devices based on ferrovalley materials, such as valley-based nonvolatile random access memory and valley filter, are contemplated for valleytronic applications.

[1] Key Laboratory of Polar Materials and Devices, Ministry of Education, East China Normal University, Shanghai, 200241, China. [2] National Laboratory of Solid State Microstructures and Department of Physics, Nanjing University, Nanjing, Jiangsu 210093, China. [3] Collaborative Innovation Center of Advanced Microstructures, Nanjing University, Nanjing, Jiangsu 210093, China. [4] Collaborative Innovation Center of Extreme Optics, Shanxi University, Taiyuan, Shanxi 030006, China. Correspondence and requests for materials should be addressed to C.-G.D. (email: cgduan@clpm.ecnu.edu.cn).

With the discovery of graphene[1], the concept of valleytronics has attracted immense attention[2,3]. Similar to charge and spin of electrons in electronics and spintronics, the valley degree of freedom in the field of valleytronics constitutes the binary states. This leads to a great deal of unconventional phenomena and possibilities for practical applications, especially in information processing industry[4,5].

Among valleytronic materials, monolayers of 2H-phase transition metal dichalcogenides (TMDs)[6–10] are the most promising ones with unique potential for utilizing and manipulating valley index effectively. With respect to centrosymmetric graphene, the space inversion symmetry for these 2H-phase TMDs is explicitly broken that gives rise to the existence of the valley Hall effect[4], as well as the valley-dependent optical selection rules[11]. Particularly, the noncentrosymmetry together with intrinsic spin–orbit coupling (SOC) derived from the $d$ orbitals of heavy transition metals[12] induce strong coupled spin and valley degree of freedom. Therefore, 2H-TMDs monolayers are generally regarded as the promising platform for studies of the fundamental physics in spintronics, valleytronics and crossing areas.

The pristine TMDs monolayers, however, are not suitable for direct information storage, as the valleys in these systems are not polarized. In analogy to paraelectric and paramagnetic materials, they can be called paravalley materials. In this regard, the major challenge in valleytronics is to break the degeneracy between the two prominent $K_+$ and $K_-$ valleys, that is, to achieve the valley polarization. At present stage, the principal mechanism invoked in the context is circularly polarized optical excitation[13–15]. However, as a dynamic process, optical pumping merely changes the chemical potential in two valleys. It does not meet the requirement of robust manipulation. Zhang et al.[3] proposed that electron–electron interaction can be an effective way to break time-reversal symmetry and then induce valley polarization. In their work, an external electric field, however, is necessary to stabilize the system. Another typical strategy through an external magnetic field[16–19], as it turns out, indeed lifts the valley degeneracy energetically. Unfortunately, the extreme field strength for a sizable valley splitting is not accessible in practical use. The approach based on optical Stark effect offers additional way to control valley-selective energy level[20]. Yet, because of the huge required amplitude of oscillating electric field, it appears to suffer from a similar problem as applying a magnetic field. More importantly, all above and many other attempts[21,22] have a fatal limitation, that is, volatility. When the applied external fields including force, electric, magnetic or optical ones are removed, the system recovers to the initial paravalley state. For the purpose of applying in next-generation electronic products with nonvolatility, scheme by means of magnetic doping[23–27] appeared as an alternative approach. In consideration of the electronic transports suffering from impurity scattering, a more intelligent way using the magnetic proximity effect[28,29] is proposed very recently. Although there exists giant and tunable valley degeneracy splitting in $MoTe_2$ induced by EuO, it is still an external method. To explore intrinsic valley polarization in TMDs is thus highly desirable.

In this study, we rebuild the Hamiltonian for the classical monolayers of TMDs, and point out that the coexistence of the SOC effect and exchange interaction of localized $d$ electrons is the sufficient condition for spontaneous valley polarization in this kind of systems. In addition to ferroelectric and ferromagnetic materials that have been routinely explored, we therefore for the first time unveil a ferrovalley material. As a new ferroic family member, its potential coupling with ferroelectric, ferromagnetic, ferroelastic and ferrotoroidic properties may provide novel physics in multiferroic field and promote technological innovation. We also predict that intriguing phenomena like anomalous valley Hall effect could occur in such system.

## Results

**Two-band $\mathbf{k} \cdot \mathbf{p}$ model including exchange interaction.** For representative monolayers of 2H-phase TMDs, such as $MoS_2$, they are in trigonal prismatic coordination ($D_{3h}$)[30,31]. The direct band gaps are located at valleys $K_+$ and $K_-$ with $C_{3h}$ point group symmetry. The bottom of the conduction band (CB) dominantly consists from $d_{z^2}$ orbitals on transition metal, involving with a minor contribution from the $p$ orbitals of chalcogens. At the top of the valence band (VB), there exists mainly hybridization between $d_{x^2-y^2}$ and $d_{xy}$ states of cation to interact with $p_x$ and $p_y$ states on anions. A two-band $\mathbf{k} \cdot \mathbf{p}$ model can be used to describe the electronic properties near the Dirac points $K_\pm$[4,9]. Note that the basis functions are chosen as $|\psi_u^\tau> = |d_{z^2}>$ and $|\psi_l^\tau> = (|d_{x^2-y^2}> + i\tau|d_{xy}>)/\sqrt{2}$ ($\tau = \pm 1$ denotes the valley index). The $p$ orbitals on the chalcogen are neglected in the model. Here, subscripts $u$ (upper band (UB)) and $l$ (lower band (LB)), instead of CB and VB, are adopted to describe the valley states. In order to violate the time inversion symmetry and induce the valley polarization, additional term $H_{ex}(\mathbf{k})$ is introduced to the effective Hamiltonian. We then construct the total Hamiltonian as follows:

$$H(\mathbf{k}) = I_2 \otimes H_0(\mathbf{k}) + H_{SOC}(\mathbf{k}) + H_{ex}(\mathbf{k}). \quad (1)$$

To reproduce the anisotropic dispersion and more importantly the electron-hole asymmetry, the first term with up to second-nearest-neighbour hopping is given by[32,33]:

$$H_0(\mathbf{k}) =$$
$$\begin{bmatrix} \frac{\Delta}{2} + \varepsilon + t_{11}'(q_x^2 + q_y^2) & t_{12}(\tau q_x - iq_y) + t_{12}'(\tau q_x + iq_y)^2 \\ t_{12}(\tau q_x + iq_y) + t_{12}'(\tau q_x - iq_y)^2 & -\frac{\Delta}{2} + \varepsilon + t_{22}'(q_x^2 + q_y^2) \end{bmatrix},$$
$$(2)$$

in which $\Delta$ is band gap at the valleys ($K_\pm$), $\varepsilon$ is a correction energy bound up with the Fermi energy, $t_{12}$ is the effective nearest neighbour hopping integral, $t_{11}'$, $t_{12}'$ and $t_{22}'$ are parameters related to the second-nearest-neighbour hopping and $q = \mathbf{k} - \mathbf{K}$ is the momentum vector. $I_2$ is the $2 \times 2$ identity matrix.

The second term, that is, the SOC term, can be written as:

$$H_{SOC}(\mathbf{k}) = \frac{\tau\lambda}{2} \begin{bmatrix} L_z & L_x - iL_y \\ L_x + iL_y & -L_z \end{bmatrix} + H'_{SOC}. \quad (3)$$

Here, $L_x$, $L_y$, $L_z$ are the $2 \times 2$ matrix for $x$, $y$, $z$ components of the orbital angular momentum. The perturbation correction $H'_{SOC}$ as a valley-dependent $4 \times 4$ matrix is applied here to incorporate the contributions from $p$ orbitals of anions and the remote $d_{xz}$ and $d_{yz}$ characters on transition metal[31]. The SOC effect directly causes the spin splitting at the bottom of the UB (the top of the LB). We label it as $2\lambda_u$ ($2\lambda_l$), defined by the energy difference $E_{u(l)\uparrow} - E_{u(l)\downarrow}$ at the $K_+$ point.

The most crucial term we imported originates from the intrinsic exchange interaction of transition-metal $d$ electrons:

$$H_{ex}(\mathbf{k}) = \sigma_z \otimes \begin{bmatrix} -m_u & 0 \\ 0 & -m_l \end{bmatrix}, \quad (4)$$

where $\sigma_z$ is the Pauli matrix, and $m_u(m_l) = E_{u(l)\downarrow} - E_{u(l)\uparrow}$ represents the effective exchange splitting in the band edge of UB (LB). The exchange interaction, equivalent to an intrinsic magnetic field, tends to split the spin-majority and spin-minority states. Combining the SOC effect that leads to coupled spin and valley indexes with the valley-independent exchange interaction, the valley polarization is therefore feasible.

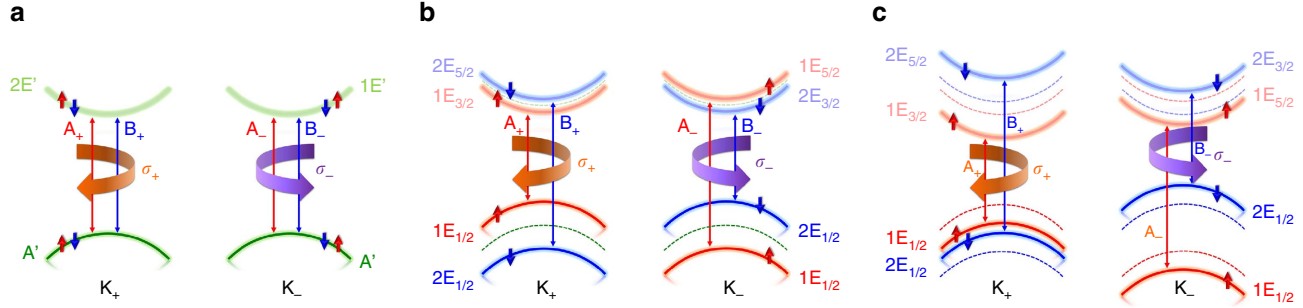

**Figure 1 | The schematic band structures at valleys $K_+$ and $K_-$ of representative 2H-phase TMD monolayers.** (**a**) Without SOC effect, (**b**) with SOC effect and (**c**) with SOC effect and a positive exchange field, that is, the valley-polarized case. The IRs of states have been labelled using the Mülliken notations. The allowed interband transitions excited by circularly polarized light near the band edges have been plotted as $A_+$, $B_+$, $A_-$ and $B_-$. $\sigma_+$ and $\sigma_-$ represent the left-handed and right-handed radiation, respectively. Note that the band structures are referenced to the one of monolayer MoS$_2$. The spin splitting of UB induced by SOC effect has the opposite sign to the one of LB. In addition, the effective exchange splitting in the band edge of UB considered here is slightly larger than that of LB.

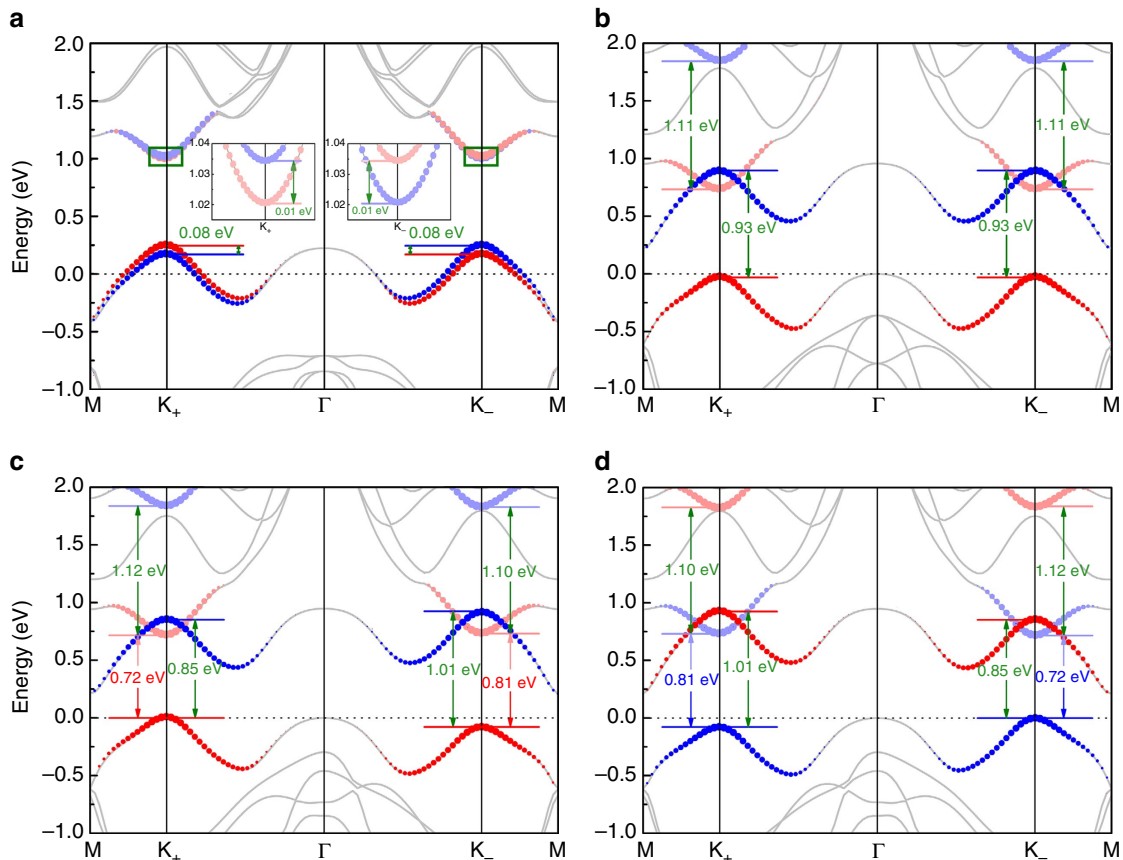

**Figure 2 | The band structures of 2H-VSe$_2$ monolayer.** (**a**) With SOC effect but without ferromagnetism, and (**b**) with magnetic moment but without SOC effect. (**c**) The real case including both the magnetism and SOC effect. (**d**) is same as (**c**) but with opposite magnetic moment. The insets in (**a**) amplify the spin splitting at the bottom of the UB. The radius of dots is proportional to its population in corresponding state near valleys: red and blue ones for spin-up and spin-down components of $d_{x^2-y^2}$ and $d_{xy}$ orbitals on cation-V, and light red and light blue symbols represent spin-up and spin-down states for $d_{z^2}$ characters. The Fermi level $E_F$ is set to zero in each cases.

**Valley-dependent optical selection rule: group theory analysis.** According to the total Hamiltonian, the band structures near the valleys $\mathbf{K}_\pm$ of classical TMDs monolayers are easily deduced. They are schematically drawn in Fig. 1, in which the Fermi level is located at the gap between UB and LB. In spite of the identical occupations, the symmetry between points $K_+$ and $K_-$ is quite different. Previous work[15] has successfully proposed the chiral absorption in valleytronic materials based on conservation of overall azimuthal quantum number. Here, using group theory

analysis, we systematically explore the valley-dependent optical selection rules, as well as the impact of valley polarization on optical properties.

As shown in Fig. 1a with absence of the SOC effect, the irreducible representations (IRs) at $K_+$ are A′ and 2E′ for LB and UB, respectively. However, for $K_-$, they become A′ and 1E′. The IRs for the bottom of UB are different between the two valleys. Note that the IRs of states are labelled in Mülliken notations. Because of the great orthogonality theorem, the electric-dipole

transition is forbidden unless the reduction of the product representation between the IRs of initial state and the incident radiation contains the representation of the final state. When the incident light is left-handed (right-handed) circularly polarized

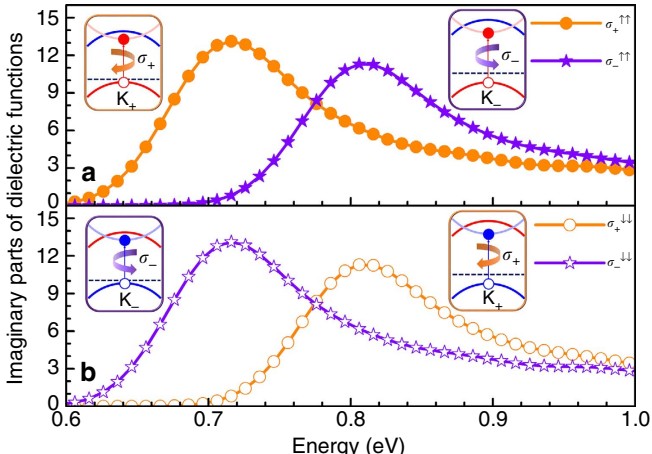

**Figure 3 | The imaginary parts of complex dielectric function $\varepsilon_2$ for monolayer VSe$_2$.** Cases excited by left-handed radiation $\sigma_+$ and right-handed radiation $\sigma_-$ (**a**) with positive magnetic moment (corresponding to Fig. 2c) and (**b**) with negative one (related to Fig. 2d) are presented. Insets are the schematic interband transitions related to certain $E_g^{opt}$.

with 2E′ (1E′) symmetry, we note that A′ $\bigotimes$ 2E′ (1E′) = 2E′ (1E′). Apparently, the optical absorption at $K_+$ ($K_-$) could only be excited by the left-handed (right-handed) light that implies the valley-dependent optical selection rules. At present, the optical band gaps ($E_g^{opt}$) are identical, that is, $E_g^{opt}(A_+) = E_g^{opt}(B_+) = E_g^{opt}(A_-) = E_g^{opt}(B_-) = \Delta$.

When the SOC effect is taken into account, the symmetry for valleys has to be interpreted by the double group $C_{3h}^D$. The identical representation A′ degenerates to 1E$_{1/2}$ and 2E$_{1/2}$ for spin-up and spin-down components. Meanwhile, 2E′ and 1E′ change to 1E$_{3/2}$ (spin-up), 2E$_{5/2}$ (spin-down) and 1E$_{5/2}$ (spin-up), 2E$_{3/2}$ (spin-down), accordingly. By applying direct product between IRs of the ground state and circularly polarized light, the allowed interband transitions can be easily obtained, as labelled in Fig. 1b. Remarkably, the chirality is locked in each valley. The valley-dependent SOC effect splits the previously degenerated A$_+$ (A$_-$) and B$_+$ (B$_-$), and makes the $E_g^{opt}$ in K$_+$ and **K**$_-$ stemming from different spin states. Nevertheless, they still bear the same value ($E_g^{opt}(A_+) = E_g^{opt}(B_-) = \Delta - \lambda_l + \lambda_u$), owing to the protection by time-reversal symmetry.

The existence of intrinsic exchange interaction (Fig. 1c) breaks the time inversion symmetry and decouples the energetically degenerated valleys, clearly elucidating the occurrence of valley polarization. It is interesting to point out that $E_g^{opt}$ excited by the left-handed radiation ($E_g^{opt}(A_+) = \Delta - \lambda_l + \lambda_u + m_l - m_u$) and the one corresponding to the right-handed light ($E_g^{opt}(B_-) = \Delta - \lambda_l + \lambda_u - m_l + m_u$) are split by the magnitude

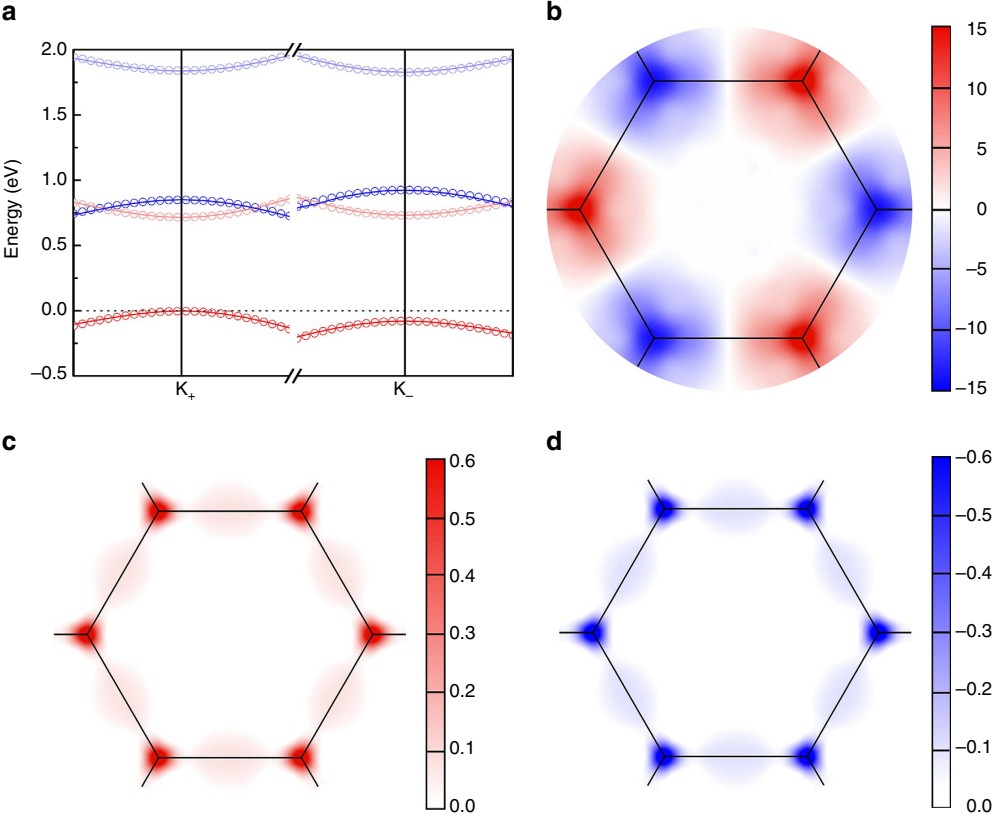

**Figure 4 | Band comparison and Berry curvatures of ferrovalley 2H-VSe$_2$ monolayer.** (**a**) Comparison of bands from the two-band $\mathbf{k} \cdot \mathbf{p}$ model (open circles) and the corresponding first-principles results (solid lines). The values of parameters in Hamiltonian are obtained via an optimal fit to the first-principles bands as the following in units of eV: $\Delta = 0.855$, $\varepsilon = 0.851$, $t_{12} = 0.343$, $t_{11}' = 0.171$, $t_{12}' = -0.174$, $t_{22}' = 0.087$, $\lambda_l = 0.038$, $\lambda_u = -0.007$ ($\lambda_l$ derives from the first term in equation (3) with $\lambda = 0.038$ eV, $L_z = $ diag $\{0,2\}$. The matrix elements of $L_x$ and $L_y$ are all zeros. $\lambda_u = \lambda'$ comes from the second term with the diagonal matrix as $H'_{SOC} = $ diag $\{\tau\lambda', 0, -\tau\lambda', 0\}$.), $m_l = 0.464$, $m_u = 0.555$. (**b**) Contour maps of Berry curvatures in the $\mathbf{k}$ space for bands mainly occupied by $d_{x^2-y^2}$ and $d_{xy}$ characters in units of Å$^2$. Summation of Berry curvatures in the point $\mathbf{k}$ and its space inversion are shown in (**c**) and (**d**) for positive and negative valley polarization cases, accordingly.

of $2|m_l - m_u|$. Amazingly, reversed chirality of the incident light sees different $E_g^{opt}$ in the valley-polarized system, indicating the possibility to judge the valley polarization utilizing noncontact and nondestructive circularly polarized optical means.

**Chirality-dependent optical band gap and Berry curvatures.** The above results establish the general rule to hunt for ferrovalley materials with spontaneous valley polarization, that is, the coexistence of the SOC effect with the intrinsic exchange interaction. Here, following the strategy, we predict a certain material: 2H-VSe$_2$ monolayer[34–36], the components of which have been already found in the R polytype VSe$_2$ bulk[37]. As a peculiar ferromagnetic semiconductor among TMDs, it possesses intrinsic magnetic moment with the magnitude of 1.01 $\mu_B$ in the V-3$d$ orbitals. The strong magnetic coupling implies remarkable exchange interaction, and then significant spontaneous valley polarization. More excitingly, on the basis of mean field theory and Heisenberg model, its estimated Curie temperature reaches up to $\sim 590$K, in accordance with the work of Pan[35]. It demonstrates that 2H-VSe$_2$ monolayer could be used in valleytronics well above room temperature. Note that the pristine 1T-phase VSe$_2$ has been widely studied[38–41]. However, with the presence of space inversion symmetry, it is definitely not a ferrovalley material. In addition, the work of Chen and colleagues[34] proved that compared with 1T monolayer, the ferrovalley 2H one is the slightly more stable phase for single-layered VSe$_2$.

When we ignore the magnetism in monolayer VSe$_2$, as shown in Fig. 2a, the band structure is essentially similar to the representative one for TMDs (Fig. 1b). Yet, it is a metal with the Fermi level passing through the states predominantly comprising $d_{x^2-y^2}$ and $d_{xy}$ orbitals on cation-V. Fortunately, the intrinsic exchange interaction of unpaired $d$ electrons, equivalent to a tremendous magnetic field, completely splits the degenerated spin-up and spin-down components of the states occupied near the Fermi level (see Fig. 2b). As a result, the system manifests ferromagnetic semiconductor with a narrow indirect band gap. Though the top of the VB is located in the $\Gamma$ point, the direct band gap remains at two valleys. The relatively small (compared with group-VI dichalcogenides) but nonnegligible SOC effect combined with the strong exchange interaction originating from intrinsic magnetic moment of V-3$d$ electrons induce valley polarization, as shown in Fig. 2c,d.

When the magnetic moment is positive (Fig. 2c), the spin splitting of states mainly occupied by $d_{x^2-y^2}$ and $d_{xy}$ orbitals equals to $|2m_l - 2\lambda_l| \sim 0.85$ eV in the valley K$_+$, and this is much smaller than $|2m_l + 2\lambda_d| \sim 1.01$ eV in the valley K$_-$. Conversely, that of primarily $d_{z^2}$ states is with a relatively greater value at the point K$_+$ ($|2m_u - 2\lambda_u| \sim 1.12$ eV) than at K$_-$ ($|2m_u + 2$

$\lambda_u| \sim 1.10$ eV) because of the opposite sign between $\lambda_u$ and $m_u$. By means of these key parameters and some others, we compare the band structures received from the Hamiltonian in equation (1) with the density-functional theory results. The excellent agreement in Fig. 4a warrants the validity of the two-band $\mathbf{k} \cdot \mathbf{p}$ model we adopted here to describe the electronic properties of valley-polarized TMDs monolayers close to the valleys.

More importantly, we analyse the band gaps and find that it is smaller at valley K$_+$ than at K$_-$ with energy difference $|2\lambda_l - 2\lambda_u| \sim 0.09$ eV that will directly reflect in the optical properties excited by circularly polarized light. Compared with the $E_g^{opt}$ related to the left-handed radiation, the right-handed one experiences a blue shift (Fig. 3a). When the magnetic moment is inverted, as clearly displayed in Fig. 2d, our interested valley polarization possess reversed polarity. As a result, in comparison with the left-handed one, the red shift of $E_g^{opt}$ excited by right-handed light happens (Fig. 3b).

Now that we have revealed the spontaneous valley polarization in monolayer VSe$_2$, it is also interesting to inspect the Berry curvature that has crucial influence on the electronic transport properties and is the kernel parameter to various Hall effects. Here, we consider the spin-resolved nonzero $z$-component Berry curvature from the Kubo formula derivation[42]:

$$\Omega_{n,z}^{\uparrow(\downarrow)}(\mathbf{k}) = -\sum_{n' \neq n} \frac{2\mathrm{Im}\left\langle \varphi_{n,\mathbf{k}}^{\uparrow(\downarrow)} \left| v_x \right| \varphi_{n',\mathbf{k}}^{\uparrow(\downarrow)} \right\rangle \left\langle \varphi_{n',\mathbf{k}}^{\uparrow(\downarrow)} \left| v_y \right| \varphi_{n,\mathbf{k}}^{\uparrow(\downarrow)} \right\rangle}{(E_{n'}^{\uparrow(\downarrow)} - E_n^{\uparrow(\downarrow)})^2}, \quad (5)$$

where $v$ is velocity operator. The Berry curvatures $(\Omega_{l,z}(\mathbf{k}) = \Omega_{l,z}^{\uparrow}(\mathbf{k}) + \Omega_{l,z}^{\downarrow}(\mathbf{k}))$ for the bands with major contribution from $d_{x^2-y^2}$ and $d_{xy}$ states of V atoms, that is, the summation between blue and red ones in Fig. 2, have been calculated. For a system with equilibrium valleys, $\Omega_{l,z}(\mathbf{k})$ is an odd function in the momentum space because of time-reversal symmetry and broken space inversion symmetry. Although the absolute values in opposite valleys are no longer identical in the ferrovalley material, Berry curvatures still have opposite sign, as displayed in Fig. 4b. We would like to emphasize that reversal of valley polarization makes the absolute values of Berry curvatures in valleys K$_+$ and K$_-$ exchanged. The sign of Berry curvature, however, stay the same. To explore the difference between $\Omega_{l,z}(\mathbf{k})$ and $\Omega_{l,z}(-\mathbf{k})$, they are summated. For the case with positive valley polarization induced by positive magnetic moment (Fig. 4c), the absolute value of Berry curvature in K$_+$ valley is greater than the one in the valley K$_-$, giving rise to a positive summation. Not surprisingly, same value with opposite sign is obtained when the valley polarization has been reversed (Fig. 4d). Consequently, Berry curvatures, as circularly polarized radiations, are another effective methods to determine the occurrence of valley polarization and its polarity reversal.

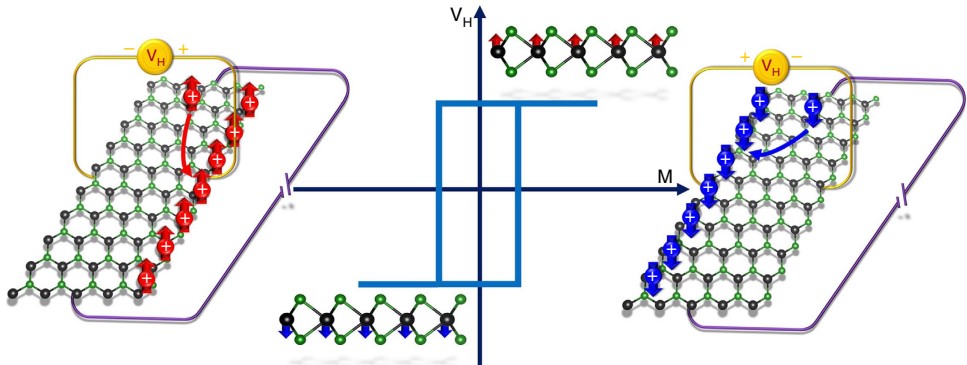

**Figure 5 | Sketch of data storage utilizing hole-doped ferrovalley materials based on anomalous valley Hall effect.** The carriers denoted by white '+' symbol are holes. Upward arrows in red colour and downward arrows in blue colour represent spin-up and spin-down carriers, respectively.

## Discussion

As we know, a direct result related to the sign change of Berry curvatures in different valleys is a new form of Hall effect, namely valley Hall effect that has been widely investigated in systems with two-dimensional honeycomb lattice[4,6,9,43–45]. Long-lived spin and valley accumulations on sample sides bring charming phenomena, such as emission of photons with opposite circular polarizations on the two boundaries. Moreover, the coexistence of spin and valley Hall current provides a route toward the integration of spintronics and valleytronics[9].

We point out that the valley Hall effect in ferrovalley materials possesses a more interesting feature, that is, the presence of additional charge Hall current originating from the spontaneous valley polarization. Analogous to the anomalous Hall effect in ferromagnetic materials, we name this effect in ferrovalley materials as anomalous valley Hall effect.

As the charge Hall current can be more easily measured in experiments, the anomalous valley Hall effect offers a possible way to realize data storage utilizing ferrovalley materials. An example for moderate hole doping VSe$_2$ with Fermi energy lying between the VB tops of $K_+$ and $K_-$ valleys is displayed in Fig. 5. It is intriguing to point out that the $p$-type VSe$_2$ possesses 100% spin polarizability around the Fermi level. Because of the almost zero Berry curvature in the centre area of Brillouin zone, the carries from the $\Gamma$ point and its neighbours pass through the ribbon directly without transverse deflection. In addition, skew scattering and other effects due to intervalley scattering are ignored here[4]. When the $p$-type VSe$_2$ possess positive valley polarization, the majority carriers, that is, spin-down holes from $K_+$ valley, gain transverse velocities towards left side in the presence of external electric field. The accumulation of holes in the left boundary of the ribbon generate a charge Hall current that can be detected as a positive voltage. When the polarity of valley reversed, spin-up holes from $K_-$ valley, as net carriers, accumulate in the right side of the sample because of the negative Berry curvature. Obviously, they lead to measurable transverse voltage with opposite sign. Note that in the anomalous valley Hall effect, there exists only one type of carrier coming from a single valley, resulting in an additional charge Hall current. The combination of valley, spin and charge accumulations implies the correlation among charge, spin and valley degree of freedom that makes the new member of Hall family absolutely different from any other forms and attractive in electronics, spintronics, valleytronics and even their crossing areas.

Based on the anomalous valley Hall effect, the electrically reading and magnetically writing memory devices are coming up. The binary information is stored by the valley polarization of the ferrovalley material that could be controlled by the magnetic moment through an external magnetic field. In addition, it can be easily 'read out' utilizing the sign of the transverse Hall voltage. Besides the nonvolatile data storage, the ferrovalley materials with spontaneous large valley polarization are ideal candidates for valley filter, valley valve and other promising valleytronic devices[5,29,46]. We strongly advocate experimental efforts on monolayer 2H-VSe$_2$ and other 2H-phase V-group dichalcogenides, where a series of ferrovalley materials are very likely to hide. It is of great importance in paving the way to the practical applications of valleytronics.

## Methods

***Ab initio* calculations.** The calculations of monolayer VSe$_2$ are performed within density-functional theory using the accurate full-potential projector augmented wave method, as implemented in the Vienna *ab initio* Simulation Package (VASP)[47]. The exchange-correlation potential is treated in the PBE (Perdew–Burke–Ernzerhof) form[48] of the generalized gradient approximation with a kinetic energy cutoff of 600 eV, as others did[34–36]. We also check that our results are qualitatively robust within the Ceperly–Alder functional form of the local density

approximation and taking the Hubbard $U$ into account to describe the on-site Coulomb repulsion between V-$d$ electrons. A $18 \times 18 \times 1$ and $36 \times 36 \times 1$ Monkhorst-Pack $k$-point mesh centred at $\Gamma$ are respectively adopted in the geometry optimization and self-consistent calculations. The convergence criterion for the electronic energy is $10^{-6}$ eV and the structures are relaxed until the Hellmann–Feynman forces on each atoms are $<1$ meV Å$^{-1}$. For the optical property calculations, we adopt our own code OPTICPACK that has been successfully applied to study the spin-dependent optical properties in ferromagnetic materials[49]. The imaginary part of the complex dielectric function $\varepsilon_2$ excited by circularly polarized light is calculated using the following relations:

$$[\varepsilon_2]_{\pm}^{\uparrow(\downarrow)}(E) = \frac{4\pi^2}{\Omega} \sum_{\mathbf{k}} W_{\mathbf{k}} \sum_{v,c} \left| p_{\pm}^{\uparrow(\downarrow)} \right|^2 \frac{\delta(E_c^{\uparrow(\downarrow)} - E_v^{\uparrow(\downarrow)} - E)}{(E_c^{\uparrow(\downarrow)} - E_v^{\uparrow(\downarrow)})^2}, \quad (6)$$

here $p_{\pm}^{\uparrow(\downarrow)}(\mathbf{k}) = (p_x^{\uparrow(\downarrow)}(\mathbf{k}) \pm i p_y^{\uparrow(\downarrow)}(\mathbf{k}))/\sqrt{2} = <\psi_{c,\mathbf{k}}^{\uparrow(\downarrow)}|p_{\pm}|\psi_{v,\mathbf{k}}^{\uparrow(\downarrow)}>$ is the electron momentum matrix element of circular polarization between the VB states ($v$) and the CB states ($c$). The momentum operator defines as $p = -im_e[r, H]/\hbar$. Note that the SOC term has been included in the effective one-electron Hamiltonian. $E$ is the photon energy, and $\Omega$ is the cell volume. The integral over the $\mathbf{k}$ space has been replaced by a summation over special $\mathbf{k}$ points with corresponding weighting factor $W_{\mathbf{k}}$. The second summation includes $v$ and $c$ states, based on the reasonable assumption that the VB is fully occupied, whereas the CB is empty. Spin-flip effects due to SOC are tiny and therefore neglected here[50].

**Data availability.** The data that support the findings of this study are available from the corresponding author on request.

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

## Acknowledgements

This work was supported by the National Key Project for Basic Research of China (Grant Nos. 2014CB921104 and 2013CB922301), the National Natural Science Foundation of China (Grant Nos. 51572085, 11525417 and 11374137) and ECNU Outstanding Doctoral Dissertation Cultivation Plan of Action (No. PY2015048). Computations were performed at the ECNU computing centre. We sincerely acknowledge useful discussions with Professor J. Feng and Professor F. Zhang.

## Author contributions

C.-G.D. conceived the idea and supervised the work. W.-Y.T. carried out the two-band **k · p** model and first-principles calculations and did the data analysis. W.-Y.T., S.-J.G. and X.-G.W. contributed to the interpretation of the results. W.-Y.T. and C.-G.D. co-wrote the paper. All the authors reviewed and modified the manuscript.

## Additional information

**Competing financial interests:** The authors declare no competing financial interests.

