## [Peer Review File · Nature Communications]

Reviewers' comments:

Reviewer #1 (Remarks to the Author):

The paper summarises the band structure peculiarities of 2H TMDs under consideration of spin-orbit coupling and external fields. Then, it applies this to 2D crystal VSe₂, which exhibits strong magnetic coupling. An anomalous Hall effect is predicted by the calculations.

VSe₂ and its magnetism has been subject to quite a few theoretical studies (e.g. doi: 10.1063/1.4893027, 10.1088/0953-8984/28/6/064002, 10.1016/j.jssc.2013.03.008, 10.1021/nn204667z). The Hall effect has been reported in a sandwich structure incorporating VSe₂ (10.1557/jmr.2015.354). The idea of applying an electric field to manipulate the split valleys was, for example, reviewed here: 10.1039/c4cs00276h.

The combination of the anomalous Hall effect with the magnetism of the system has not been discussed before. The question is if this is such surprising result to merit a Nature Comm., or is this incremental. In particular, is the Hall effect of the same origin as the one published in the sandwich (10.1557/jmr.2015.354). This requires more analysis and discussion.

The paper is not written in the required separation of results and discussion. A discussion is not existing as such, just a concluding paragraph. On the other hand, what is called "results" is not a section that lives solely on the results of the authors. The paper should be reorganised.

Scientifically the paper appears to be sound and everything is well justified.

I believe the strong point of the paper (which is beyond Figure 2) should be made very clear. Results and discussion must really be separated. References given above should be cited. Most importantly, it must be made clear if this paper really is a strong new contribution to the 2D QSH field.

Reviewer #2 (Remarks to the Author):

The authors propose to investigate a interesting 2D material, namely 2H-VSe₂ having localized magnetic moment on V atoms. The idea is to combine valleytronics and magnetism. The approach seems original and add new informations and results in the whole valleytronic panorama. They propose to extend the concept of ferro-(magnetism/electricity) to ferro-(valley) and introduce the idea of ferrovalley materials, where the exchange splitting due the magnetic atoms remove the degeneracy of K⁺ and K⁻ valley. This lead to potentially interesting properties.

The authors support their idea by building a reasonable tight binding model which provide preliminary informative results and properties which are later benchmarked by quantitative calculations.

The results and the new properties appear to be well discussed and justified. Therefore I'm in favor for publication of this work in Nat. Comm.

Minor issues:

the English should be improved; I would avoid terms like "star of outlook", "following the tactic", and similar.

row 39: remove-removed!?

row 24: declaring !?
row 27: thankfully ?!
row 79: inversed (reverted)
row 113: DiscussionConclusion.

Reviewer #3 (Remarks to the Author):

Authors propose the concept of "ferrovalley material" and "anomalous valley Hall effect" within the context of functional devices based on valleytronic effects. The manuscript is not easy to read due to the English form, that should be revised and checked for typos, long sentences and expressions not proper of the standard scientific English.

The simplicity of the proposed effective hamiltonian is the strength of authors' manuscript. They show in an elegant way that the exchange interaction combined with SOC is the key to obtain a valley polarization. The effective hamiltonian is properly explained and correctly connected with the results discussed in the paper. The related results could constitute a significant advance in the field if they were supported by experimental evidences. If authors cannot provide experimental results supporting their theoretical work, maybe they can find examples in literatures that can be explained by their model.

Despite the importance of their work, in the present form and in the absence of experimental data supporting authors' conclusions, I believe that the manuscript is not suitable for publication in Nature Communications; instead, it would perfectly fit the scope of more specialized journals such as Physical Review B. If authors are able to provide experimental data supporting the conclusions and they address the issues outlined below, I encourage them to resubmit the revised manuscript to Nature Communications with a more extended discussion including the new data and outcomes.

Other issues should be also addressed in order to make the manuscript suitable for publishing in any journal:

- 1) In "Methods" section, authors should justify the choice of the PBE functional and state what is the k-mesh used to sample the Brillouin zone. Moreover, authors should explain the meaning of all the variables in equation 6: even if authors use a standard notation, this is necessary for the sake of clarity.
- 2) Line 16-18: authors speak suddenly about an effect specific of the hexagonal symmetry without introducing why this is needed. They mention in the abstract that the study will focus on 2H-VSe₂ that has hexagonal symmetry but this should be properly introduced in the main text for the sake of fluency and clarity of the arguments discussed in the introductory paragraph.
- 3) Line 23, "the space inversion symmetry for these 2D materials are explicitly broken": if I understood the grammar mistake correctly, I guess authors refer to the fact that the inversion symmetry is broken in monolayers with respect to the bulk counterpart, but this should be explicitly said.
- 4) Line 89. "While for K...changes as 1E": sentence is not complete, thus not clear.

5) Authors should provide more examples than only the 2H-VSe₂ monolayer: in similar materials, the strong hybridization with the p states of the chalcogen atom could play a fundamental role in the proposed valleytronic effects. In this respect, authors should show the atom-projected density of states and discuss what is the role of the d-p hybridization, and if it is significant. I would expect that their conclusion would be quite different for systems like WSe₂, where the wider spread of the d-like electronic density of the transition metal would produce a higher hybridization with the p orbitals of the chalcogen atom, the latter thus playing a fundamental role in the proposed valleytronic effects.

6) As a personal suggestion, when authors opt for the "double blind" peer review option, they should remove or modify any sentence in the manuscript that could unveil authors' identity, as suggested in Nature authors' guidelines. For example, in the method section, they say "For the optical property calculations, we adopt our own code OPTICPACK..." providing the reference. Furthermore, if authors want to preserve their anonymity during the peer review process, they should not upload the full version in the arXiv server (see <https://arxiv.org/abs/1604.05833>).

REVIEWERS' COMMENTS:

Reviewer #1 (Remarks to the Author):

I am very happy with the revision of the manuscript. I think my questions, partially due to fast reading, have improved the presentation a bit.

I would just like the authors to add a little discussion about their material. They write now that two layered phases (T and H) should exist, but a little more information to the reader who is unfamiliar with VSe₂ is needed. The discussion should include (i) stability, in particular inbetween the phases (see e.g. the reference to Zhongfang Chen's work), (ii) potential availability - I found some references to experimental VSe₂, though not to 2H VSe₂ (possibly due to lack of resources as I am travelling).

Thus, after this point is addressed (minor revision) I recommend acceptance.

Reviewer #3 (Remarks to the Author):

I went through the revised version of the manuscript and I believe that authors made their best effort to answer the reviewers' questions. The work is improved and properly justified. However, a direct experimental application of their formulation is lacking. Despite authors provided examples in their response, those are not quantitative and not directly tied with the quantities present in their formulation. For this reason I cannot still recommend the publication of their work in Nature Communications. Nevertheless, the power of their method and the immediate interest of other experimental groups demonstrate the importance of authors' work. As a such, I renew my suggestion to submit their work to other journals like Physical Review B, or similar; if authors are able to prove more extensively the general result of their work, I also suggest to submit their manuscript to journal of wider audience like Scientific Report.

Response to the report (NCOMMS-16-09754)

----- Reply to the Reviewer #1 -----

The paper summarises the band structure peculiarities of 2H TMDs under consideration of spin-orbit coupling and external fields. Then, it applies this to 2D crystal VSe₂, which exhibits strong magnetic coupling. An anomalous Hall effect is predicted by the calculations.

VSe₂ and its magnetism has been subject to quite a few theoretical studies (e.g. doi: 10.1063/1.4893027, 10.1088/0953-8984/28/6/064002, 10.1016/j.jssc.2013.03.008, 10.1021/nn204667z). The Hall effect has been reported in a sandwich structure incorporating VSe₂ (10.1557/jmr.2015.354). The idea of applying an electric field to manipulate the split valleys was, for example, reviewed here: 10.1039/c4cs00276h. The combination of the anomalous Hall effect with the magnetism of the system has not been discussed before. The question is if this is such surprising result to merit a Nature Comm., or is this incremental. In particular, is the Hall effect of the same origin as the one published in the sandwich (10.1557/jmr.2015.354). This requires more analysis and discussion.

Answer: We thank you very much for pointing out these important references to us. We have added them in the revised manuscript as Ref.[10]: *Chem. Soc. Rev.* **44**, 2603-2614 (2015); Ref.[36]: *J. Phys.: Condens. Matter* **28**, 064002 (2016); Ref.[37]: *ACS Nano* **6**, 1695-1701 (2012); Ref.[38]: *J. Solid State Chem.* **202**, 128-133 (2013); Ref.[39]: *Appl. Phys. Lett.* **105**, 063109 (2014); and Ref.[40]: *J. Mater. Res.* **31**, 886-892 (2016).

Firstly, we highly agree that applying an electric field based on valley-selective optical Stark effect (Ref.[20]: *Nat. Mater.* **14**, 290-294 (2015).) and an external magnetic field (Ref. [16]: *Nat. Phys.* **11**, 141-147 (2015); Ref. [17]: *Nat. Phys.* **11**, 148-152 (2015); Ref. [18]: *Phys. Rev. Lett.* **114**, 037401 (2015); and Ref. [19]: *Phys. Rev. Lett.* **113**, 266804 (2014).) are effective approaches to “manipulate the split valleys”. Unfortunately, there

exist the inevitable limitation, i.e. **volatility**. As mentioned in the “Introduction” part of our manuscript, when the applied external fields are removed, the valleys locked by time-reversal symmetry are still degenerate, stabilizing the system in the initial paravalley state. While, “*under consideration of spin-orbit coupling and*” the **intrinsic exchange interaction** of transition-metal-*d* electrons, we propose a way to explore **spontaneous valley polarization** in our work. As a semiconductor with both the SOC effect and “*strong magnetic coupling*”, 2H-VSe₂ is a certain material, possessing **spontaneous valley polarization independent on any external fields**, which is so-called a **ferrovalley material**.

Due to the advantage of **nonvolatility**, our strategy is apparently distinct from “*the idea of applying an electric/magnetic field to manipulate the split valleys*” reviewed in the paper you recommended “The electronic structure calculations of two-dimensional transition-metal dichalcogenides in the presence of external electric and magnetic fields” (doi: [10.1039/c4cs00276h](https://doi.org/10.1039/c4cs00276h)), as well as some other **volatile** approaches utilizing external fields including force and optical ones. In fact, the spontaneous valley polarization **in the absence of any external fields** is exactly the highlight of our work. It implies the potential application of **ferrovalley materials** in **nonvolatile** data storage and other next-generation electronic products. In analogy to ferroelectric materials with spontaneous charge polarization in electronics, as well as ferromagnetic materials with spontaneous spin polarization in spintronics, the concept of “ferrovalley material” with spontaneous valley polarization is a new member of ferroic-family, and is of both fundamentally physical and practically technological importance in ferroic and valleytronic fields.

In addition, we respectfully point out that most of the papers you recommended (doi: [10.1063/1.4893027](https://doi.org/10.1063/1.4893027), [10.1016/j.jssc.2013.03.008](https://doi.org/10.1016/j.jssc.2013.03.008), [10.1021/nn204667z](https://doi.org/10.1021/nn204667z), and [10.1557/jmr.2015.354](https://doi.org/10.1557/jmr.2015.354)) focused on **1T-phase VSe₂**. Fig. 1(a) in “Evidence of the existence of magnetism in pristine VX₂ monolayers (X = S, Se) and their strain-induced tunable magnetic properties” (doi: [10.1021/nn204667z](https://doi.org/10.1021/nn204667z)), the right panel of Fig. 1 in “Synthesis, structure and electrical properties of a new tin vanadium selenide” (doi: [10.1016/j.jssc.2013.03.008](https://doi.org/10.1016/j.jssc.2013.03.008)), and the inset of Fig. 2 in “Transport properties of VSe₂

monolayers separated by bilayers of BiSe” (doi: [10.1557/jmr.2015.354](https://doi.org/10.1557/jmr.2015.354)) clearly show the geometric structure of 1T-VSe₂, characterized by space inversion symmetry. In the paper “Thickness dependence of the charge-density-wave transition temperature in VSe₂” (doi: [10.1063/1.4893027](https://doi.org/10.1063/1.4893027)), the metallic behavior as shown in Fig. 3(a) demonstrates that the VSe₂ samples are in 1T-phase. With the presence of the space inversion symmetry and metallic behavior, 1T-VSe₂ is definitely not a ferrovalley material. The structural difference between 1T-VSe₂ and the ferrovalley 2H-VSe₂ is explicitly shown above in Fig. R1.

Figure. R1. Geometric structures (top and side views) of 2D single-layer VSe₂ in the T (a) and H phases (b). (From Ref. [34].: Li, F., Tu K. & Chen Z. Versatile electronic properties of VSe₂ bulk, few-Layers, monolayer, nanoribbons, and nanotubes: A computational exploration. *J. Phys. Chem. C* **118**, 21264-21274 (2014).)

Since the 1T-phase VSe₂ is a **ferromagnetic metal**, the Hall effect in it is surely the normal “anomalous Hall effect”. We have to emphasize that the **anomalous valley Hall effect** we introduced in ferrovalley materials is totally different from the normal “anomalous Hall effect”. The normal one, which has been widely studied since its discovery in ferromagnetic metals in 1881 (Philos. Mag. **12**, 157 (1881).), represents **only the charge Hall current**. While, the “anomalous valley Hall effect” demonstrates **the**

coexistence of charge, spin and valley Hall currents.

As known to all, the charge accumulations on sample sides can be easily measured by the Hall voltage. The detections of the spin and valley accumulations are also experimentally feasible. Awschalom and his colleagues (*Science* **306**, 1910 (2004).) successfully imaged spin accumulations based on magneto-optical Kerr effect. The valley accumulations were also investigated using polarization-resolved photoluminescence by Wu et al. (*Nat. Phys.* **9**, 149 (2013).). In addition to the experimental detectability, long-lived charge, spin and valley accumulations on sample sides bring charming phenomena, which are unavailable in the normal “anomalous Hall effect”, such as emission of photons with opposite circular polarizations on the two boundaries. Due to the coexistence among the charge, spin and valley degrees of freedom, our “anomalous valley Hall effect” provides a route towards the integration of electronics, spintronics, and valleytronics. It, undoubtedly, cannot be realized in any other forms of the Hall family. As mentioned above, the “anomalous valley Hall effect” in 2H-VSe₂ in our work is **not** “of the same origin as the one published in the sandwich (10.1557/jmr.2015.354)”.

In order to clearly demonstrate the difference of VSe₂ between the 1T- and 2H-phase and avoid possible misunderstanding, the sentence “Noted that the pristine 1T-phase VSe₂ has been widely studied³⁷⁻⁴⁰. However, with the presence of space inversion symmetry, it is not a ferrovalley material.” has been added in the first paragraph of the subsection “Chirality-dependent optical band gap and Berry curvatures in the ferrovalley material”.

The paper is not written in the required separation of results and discussion. A discussion is not existing as such, just a concluding paragraph. On the other hand, what is called "results" is not a section that lives solely on the results of the authors. The paper should be reorganised.

Answer: We apologize that the main text of our previous manuscript was not divided appropriately. According to your suggestion and the “Guide to authors” of Nature

Communications, we seriously analyze the structure of the main text. Now, it is reorganized in the revised version. The origin subsection “Anomalous valley Hall effect” now becomes “Discussion”. The concluding paragraph is removed, whose contents accordingly transfer to proper places in the main text. More details are available in the “Summary of Changes” (see below). We believe that the sections headed “Results” and “Discussion” in the resubmitted manuscript are really separated this time.

Scientifically the paper appears to be sound and everything is well justified.

Answer: We sincerely thank you for your positive comment of our work.

I believe the strong point of the paper (which is beyond Figure 2) should be made very clear. Results and discussion must really be separated. References given above should be cited. Most importantly, it must be made clear if this paper really is a strong new contribution to the 2D QSH field.

Answer: Following your suggestions, results and discussion sections have been already really separated. All the references you recommended have been cited in our revised manuscript, as summarized in the “Summary of Changes” (see below).

As displayed in Fig. R2, the “Hall effect”, discovered by Hall in 1879 (Am. J. Math **2**, 287 (1879).), represents the charge Hall current under external magnetic field. The “anomalous Hall effect”, as discussed above, signifies a similar charge accumulations in the absence of magnetic field. The “spin Hall effect” (Phys. Lett. A **35**, 459 (1971).) introduces the spin index of electrons into the Hall family. All of the above correspond to the single type of Hall current.

A New Form of Hall Effect: Anomalous Valley Hall Effect

Figure. R2. Schematic pictures of various forms of Hall effect.

With the birth and development of valleytronics, the concept of “valley Hall effect” (Phys. Rev. Lett. **99**, 236809 (2007).), originating from the sign change of Berry curvatures in different valleys, emerged. There exists combination between spin and valley Hall current. In this work, we propose a new member of ferroic-family, i.e. a **ferrovalley material** with **spontaneous** valley polarization. Due to the different absolute value between the Berry curvatures in inequivalent valleys, an additional charge Hall current is present, except for the spin and valley accumulations in the normal “valley Hall effect”. It makes the Hall effect in ferrovalley materials quite special. Analogous to the anomalous Hall effect in ferromagnetic materials (corresponding to the normal Hall effect), we name this effect in ferrovalley materials as “**anomalous valley Hall effect**”. As a **new form of Hall effect**, it is characterized by **the coexistence of charge, spin and valley Hall current**, and thus is totally different from any other ones. More excitingly, the long-lived charge, spin and valley accumulations on sample sides is **independent to any external fields** and can be experimentally detected as mentioned above.

From the perspective of fundamental physics, it not only implies the correlation among charge, spin and valley degrees of freedom, but also brings charming phenomena, such as 100% polarized spin currents and emission of photons with circular polarizations on the two boundaries. From the point of view of practical applications, it reveals the possibility of utilizing valleytronic materials in **nonvolatile** data storage and other next-generation electronic products. More importantly, it provides a route towards the integration of electronics, spintronics and valleytronics. Based on the above, we are confident that our work is “*really a strong new contribution to*” both 2D valleytronic and Hall effect fields.

Based on your suggestions, some changes are adopted to highlight our strong points. For example, the sentence “*The combination of valley, spin and charge accumulations implies the correlation among charge, spin and valley degrees of freedom, which makes the new member of Hall family absolutely different from any other forms and attractive in electronics, spintronics, valleytronics and even their crossing areas.*” is added to the section “Conclusion” in our new version.

Hope our earnest responses, listed above, appropriately answer all your concerns. And, the current manuscript fulfills your requirement.

Reply to the Reviewer #2

The authors propose to investigate a interesting 2D material, namely 2H-VSe₂ having localized magnetic moment on V atoms. The idea is to combine valleytronics and magnetism. The approach seems original and add new informations and results in the whole valleytronic panorama. They propose to extend the concept of ferro-(magnetism/electricity) to ferro-(valley) and introduce the idea of ferrovalley materials, where the exchange splitting due the magnetic atoms remove the degeneracy of K⁺ and K⁻ valley. This lead to potentially interesting properties.

The authors support their idea by building a reasonable tight binding model which provide preliminary informative results and properties which are later benchmarked by quantitative calculations.

The results and the new properties appear to be well discussed and justified. Therefore I'm in favor for publication of this work in Nat. Comm.

Answer: We thank you very much for your appreciation of our work, especially your positive recommendation and helpful advice. All the issues are addressed in the following paragraphs.

1. *the English should be improved; I would avoid terms like "star of outlook", "following the tactic", and similar.*

Answer: The phrases are corrected in our revised manuscript. “the star of outlook” is changed to “the most promising ones”. “following the **tactic**” is replaced by “following the **strategy**” . Some other long sentences and expressions not proper of the standard scientific English are also corrected, as summarized in the “Summary of Changes” (see below).

2. row 39: *remove-removed!?*

Answer: The sentence “When the applied external fields including force, electric, magnetic and optical ones **remove** ...” is corrected as “When the applied external fields including force, electric, magnetic and optical ones **are removed**...”.

3. row 124: *declaring!?*

Answer: The sentence “...**declaring** that it could be used in valleytronics well above room temperature.” is changed to “It **demonstrates** that 2H-VSe₂ monolayer could be used in valleytronics well above room temperature.”.

4. row 127: *thankfully!?*

Answer: The sentence “**Thankfully**, the intrinsic exchange interaction of unpaired d electrons,...” is replaced by “**Fortunately**, the intrinsic exchange interaction of unpaired d electrons,...”.

5. row 179: *inversed (reverted)*

Answer: We think the word “reversed” is more appropriate to be used here. So the sentence “Not surprisingly, same value with opposite sign is obtained when the valley polarization has been **inversed** (Fig. 4d).” is corrected as “Not surprisingly, same value with opposite sign is obtained when the valley polarization has been **reversed** (Fig. 4d).”. In addition, the word “inversed” of the sentence “Amazingly, **inversed** chirality of the incident light sees different...” in the last paragraph of the subsection “Valley-dependent optical selection rule based on group theory analysis” is also replaced by “reversed”.

6. row 213: *DiscussionConclusion.*

Answer: We are sorry to point out that the section headed Conclusion seems unable to be included according to the “Guide to authors” of nature communications. As mentioned by the Reviewer #1, the main text of our previous manuscript was not divided appropriately. In the revised version, it is reorganized.

Reply to the Reviewer #3

Authors propose the concept of "ferrovalley material" and "anomalous valley Hall effect" within the context of functional devices based on valleytronic effects. The manuscript is not easy to read due to the English form, that should be revised and checked for typos, long sentences and expressions not proper of the standard scientific English.

Answer: We apologize to bother you with the English form. We have carefully checked our manuscript, and tried our best to avoid typos, improper long sentences and expressions in the revised version. For example, the phrase “the star of outlook” is changed to “the most promising ones”. The long sentence “Particularly, the noncentrosymmetry together with intrinsic spin-orbit coupling (SOC) originated from the *d*-orbitals of heavy transition metals induce strong coupled spin and valley degree of freedom, making them as a promising platform for the study of the fundamental physics in spintronics, valleytronics and crossing areas.” is divided into “Particularly, the noncentrosymmetry together with intrinsic spin-orbit coupling (SOC) derived from the *d*-orbitals of heavy transition metals induce strong coupled spin and valley degree of freedom. Therefore, 2H-TMDs monolayers are generally regarded as the promising platform for studies of the fundamental physics in spintronics, valleytronics and crossing areas.”. More Details are available in the “Summary of Changes” (see below).

The simplicity of the proposed effective hamiltonian is the strength of authors' manuscript. They show in an elegant way that the exchange interaction combined with SOC is the key to obtain a valley polarization. The effective hamiltonian is properly explained and correctly connected with the results discussed in the paper. The related results could constitute a significant advance in the field if they were supported by experimental evidences. If authors cannot provide experimental results supporting their theoretical work, maybe they can find examples in literatures that can be explained by their model.

Answer: We sincerely appreciate your positive remarks, which certainly confirm the novelty and importance of our work. Your concerns about experimental proof will be seriously discussed below.

Despite the importance of their work, in the present form and in the absence of experimental data supporting authors' conclusions, I believe that the manuscript is not suitable for publication in Nature Communications; instead, it would perfectly fit the scope of more specialized journals such as Physical Review B. If authors are able to provide experimental data supporting the conclusions and they address the issues outlined below, I encourage them to resubmit the revised manuscript to Nature Communications with a more extended discussion including the new data and outcomes.

Answer: To our understanding, the reason for your decision, i.e. “*the manuscript is not suitable for publication in Nature Communications*”, mainly roots in “*the absence of experimental data supporting authors' conclusions*”. We have to admit that experimental results supporting our theoretical work cannot be provided in a very short period of time. Nevertheless, we are very pleased to see that you also offered an alternative solution as “*If authors cannot provide experimental results supporting their theoretical work, maybe they can find examples in literatures that can be explained by their model.*”. Fortunately, we indeed find a great amount of experimental evidences to prove the feasibility of our findings. We would like to demonstrate all the details in three parts, as listed in the following pages.

Firstly, compared to the classical Hamiltonian for TMDs monolayers, we additionally introduce the exchange interaction term $H_{\text{ex}}(\mathbf{k})$ to the effective Hamiltonian, in order to violate the time inversion symmetry and then induce the valley polarization. Is the strategy utilizing the coexistence between the SOC effect and the intrinsic exchange interaction, really able to achieve the valley polarization?

As we know, the exchange interaction, equivalent to an **intrinsic magnetic field**, tends to split the spin-majority and spin-minority states. In the case, if the degeneracy between \mathbf{K}_+ and \mathbf{K}_- valleys in TMDs monolayers can be broken by an **external magnetic field**, our strategy must be effectively. As mentioned in the section “Introduction” of our

manuscript, several experimental studies proved that in the presence of an external magnetic field, the valley degeneracy is indeed lifted energetically in TMDs monolayers, such as WSe₂ and MoSe₂. Details of these studies can be found in the Ref. [16]: *Nat. Phys.* **11**, 141-147 (2015); Ref. [17]: *Nat. Phys.* **11**, 148-152 (2015); Ref. [18]: *Phys. Rev. Lett.* **114**, 037401 (2015); and Ref. [19]: *Phys. Rev. Lett.* **113**, 266804 (2014). All of these experimental results imply the feasibility of our reconstructed Hamiltonian to realize the valley polarization.

We also want to point out that the extreme magnetic field strength for a sizable valley splitting is not accessible in practical use. In addition, the route based on an external magnetic field is **volatile**. When it is removed, the valleys locked by time-reversal symmetry are still degenerate, stabilizing the system in the initial paravalley state. Here, we propose a **nonvolatile** strategy to realize **spontaneous** valley polarization. In general, the intrinsic exchange interaction is equivalent to a tremendous magnetic field $\sim 10^4$ T, which is far beyond the external field accessible in the laboratory and is huge enough to completely split the degenerated valleys. Noted that the spontaneous valley polarization **in the absence of any external fields** is different from external approaches and exactly is the highlight of our work. It implies the potential application of ferrovalley materials in **nonvolatile** data storage and other next-generation electronic products.

Secondly, is 2H-VSe₂ monolayer a certain ferrovalley material with spontaneous valley polarization? The principal difference between the ferrovalley material and the normal paravalley one, such as representative MoS₂, roots in the existence of intrinsic exchange interaction, or equivalently magnetism. Although, single-layer 2H-VSe₂ sheet has not been experimentally realized so far, the ferromagnetism of 1T-VSe₂ bulk crystal has been extensively studied (*Phys. Rev. Lett.* **109**, 086401 (2012); *Phys. Rev. B* **82**, 075130 (2010).). Meanwhile, experimental work for the 1T-VSe₂ thin films demonstrated that magnetic characters of 3D bulk phase can be retained in nanostructures (*Angew. Chem.* **52**, 10477 (2013); and the references pointed out by the Reviewer #1: Ref.[39]: *Appl. Phys. Lett.* **105**, 063109 (2014); Ref.[40]: *J. Mater. Res.* **31**, 886-892 (2016).). The structural difference between 1T- and 2H-phase of VSe₂ roots in the position of anions (see the geometric structures shown in Fig. R3), while the exchange interaction in VSe₂

corresponds to the magnetic coupling between V-3d orbitals. Therefore, a similar ferromagnetic character is very likely to be found in 2H-VSe₂ as its 1T-phase possesses.

Figure. R3. Geometric structures (top and side views) of 2D single-layer VSe₂ in the T (a) and H phases (b). The gray and khaki atoms represent V and Se atoms, respectively. (From Ref. [34]: Li, F., Tu K. & Chen Z. Versatile electronic properties of VSe₂ bulk, few-Layers, monolayer, nanoribbons, and nanotubes: A computational exploration. *J. Phys. Chem. C* **118**, 21264-21274 (2014).)

In addition, Spiecker et al. (*Phys. Rev. Lett.* **96**, 086401 (2006).) observed T to R polytype transformation in the thin VSe₂ surface layer, which was induced by Cu deposition. Such transformations were also found when electrochemically intercalating Cu into bulk VSe₂ (K. Sollmann. Ph.D. Thesis, Technical University of Berlin, 1995) or intercalating K into VSe₂ in vacuum (*Surf. Sci.* **461**, 137 (2000)). In each single-layer component of the R polytype, the V atoms are in trigonal prismatic coordination with Se atoms, representing a 2H-type monolayer.

Based on the above experimental evidences, we are confident that 2H-VSe₂ monolayer must be a certain ferrovalley material with spontaneous valley polarization, which can exist stably.

Thirdly, is it possible to measure the anomalous valley Hall effect and distinguish it from other forms of Hall effect? The “Hall effect”, discovered by Hall in 1879 (*Am. J. Math* **2**, 287 (1879).), represents the charge Hall current under external magnetic field. The “anomalous Hall effect” signifies the charge Hall current in the absence of magnetic field. The “spin Hall effect” (*Phys. Lett. A* **35**, 459 (1971).) introduces the spin index of electrons into the Hall family. All of the above correspond to the single type of Hall current. Even for the “valley Hall effect” (*Phys. Rev. Lett.* **99**, 236809 (2007).), there exists merely spin and valley Hall current. While, the “anomalous valley Hall effect”, as a new form of the Hall effect introduced by us, is characterized by **the coexistence of charge, spin and valley Hall current**.

As known to all, the charge accumulations on sample sides can be easily measured by the Hall voltage. The detections of the long-lived spin and valley accumulations are also experimentally feasible. Awschalom and his colleagues (*Science* **306**, 1910 (2004).) successfully imaged spin accumulations based on magneto-optical Kerr effect. The valley accumulations were also investigated using polarization-resolved photoluminescence by Wu et al. (*Nat. Phys.* **9**, 149 (2013).). When all of the long-lived charge, spin and valley accumulations on sample sides are experimentally observed, the sample must be a ferrovalley material with anomalous valley Hall effect.

Overall, the experimental results mentioned above are powerful evidences to prove the feasibility of our findings. In fact, related attempts are carrying out by our experimental partners. When we realized our manuscript on arXiv, several theoretical and experimental groups, such as F. Zhang’s group in the University of Texas at Dallas and C. S. Tian’s group in Fudan University, contacted with us and expressed their interests. We are confident that our findings will be supported by experimental work in the near future. Nevertheless, considering that our work are “*a significant advance*” in the highly competitive valleytronic field, we hope it can be published in Nature Communications as soon as possible, even in the absence of directly experimental data currently. Our study, which is of both fundamentally physical and practically technological importance in

spintronics, valleytronics and crossing areas, will certainly appeal to a broader audience of Nature Communications and drive experimental efforts on monolayer 2H-VSe₂ and other 2H-phase V-group dichalcogenides, where a series of ferrovalley materials are very likely to hide.

Other issues you pointed out are addressed in the following paragraphs.

1) In "Methods" section, authors should justify the choice of the PBE functional and state what is the k -mesh used to sample the Brillouin zone. Moreover, authors should explain the meaning of all the variables in equation 6: even if authors use a standard notation, this is necessary for the sake of clarity.

Answer: Since the PBE form of exchange-correlation potential has been widely used to explore the electronic properties of both 2H- and 1T-phase VSe₂ (Ref. [34].: *J. Phys. Chem. C* **118**, 21264-21274 (2014); Ref. [35].: *J. Phys. Chem. C* **118**, 13248-13253 (2014); Ref. [36].: *J. Phys.: Condens. Matter* **28**, 064002 (2016); Ref. [37].: *ACS Nano* **6**, 1695-1701 (2012).), we choose the same scheme in our work. We also reproduce our calculations utilizing the Ceperly-Alder functional form of the local density approximation (LDA) and taking the Hubbard U into account to describe the on-site Coulomb repulsion between V- d electrons. The robust results, especially the ferromagnetic semiconducting behavior of 2H-VSe₂, guarantee that our choice for the PBE form is reasonable.

Following your suggestions, the choice of the PBE functional is justified as “The exchange-correlation potential is treated in Perdew-Burke-Ernzerhof (PBE) form⁴⁸ of the generalized gradient approximation (GGA) with a kinetic-energy cutoff of 600 eV, as others did³⁴⁻³⁶. We also check that our results are qualitatively robust within the Ceperly-Alder functional form of the local density approximation (LDA) and taking the Hubbard U into account to describe the on-site Coulomb repulsion between V- d electrons”. The k -mesh used to sample the Brillouin zone is now stated in our revised manuscript as “A

18×18×1 and 36×36×1 Monkhorst-Pack k -point mesh centered at Γ are respectively adopted in the geometry optimization and self-consistent calculations.”. In addition, the meaning of the variables W_k , ν and c in equation 6, which was not demonstrated in the previous version, is clearly explained as “The integral over the k space has been replaced by a summation over special k points with corresponding weighting factor W_k . The second summation includes ν and c states, based on the reasonable assumption that the VB is fully occupied, while the CB is empty.”.

2) Line 16-18: authors speak suddenly about an effect specific of the hexagonal symmetry without introducing why this is needed. They mention in the abstract that the study will focus on 2H-VSe2 that has hexagonal symmetry but this should be properly introduced in the main text for the sake of fluency and clarity of the arguments discussed in the introductory paragraph.

Answer: From the perspective of valleytronics, the hexagonal symmetry is not necessary. In fact, very recent studies show that valley physics can also be observed in two-dimensional group-IV monochalcogenides with orthorhombic structure (Phys. Rev. B **93**, 045431 (2016); Phys. Rev. B **92**, 085406 (2015).). Nevertheless, 2H-phase TMDs with hexagonal symmetry are one of the most promising groups among valleytronic materials due to the intriguing phenomena in them, such as the valley-dependent optical selection rules, and the valley Hall effect. In order to avoid the possible misunderstanding, the expressions, such as “the hexagonal symmetry” and/or “honeycomb lattice symmetry”, are removed in our revised version. More Details are available in the “Summary of Changes” (see below).

3) Line 23, "the space inversion symmetry for these 2D materials are explicitly broken": if I understood the grammar mistake correctly, I guess authors refer to the fact that the inversion symmetry is broken in monolayers with respect to the bulk counterpart, but this should be explicitly said.

Answer: We apologize for the misunderstanding here. We refer to the fact that unlike graphene with space inversion symmetry, inversion symmetry is explicitly broken in these TMDs monolayers. As known to all, in graphene, the unit cell consists of two carbon atoms, i.e. the A and B sublattices. However, the equivalence between the two sublattices makes graphene centrosymmetric. In order to induce band gap and make practical use of valley index, various approaches have been adopted to break the inversion symmetry in graphene, such as epitaxial engineering (Nat. Mater. **6**, 770 (2007).), and interlayer voltage in biased graphene bilayer (Science **313**, 951 (2006).). Fortunately, in pristine 2H-TMDs monolayers, the inversion symmetry is intrinsically broken, leading to intriguing phenomena and the possibility to utilize and manipulate valley degree of freedom directly.

According to your suggestion, the sentence is now corrected as “With respect to centrosymmetric graphene, the space inversion symmetry for these 2H-phase TMDs is explicitly broken, which gives rise to the existence of the valley Hall effect, as well as the valley-dependent optical selection rules.”

4) Line 89. *"While for K ...changes as $1E'$ ": sentence is not complete, thus not clear.*

Answer: The sentence “While for K , the one for the bottom of the UB changes as $1E'$.” is replaced by “While for K , they become A' and $1E'$. It is obvious that the IRs for the bottom of UB are different between the two valleys.” in the reversed manuscript. We think it clearly demonstrates the symmetry for LB and UB at valley K , which is schematically displayed in Fig. 1a.

5) *Authors should provide more examples that only the 2H-VSe₂ monolayer: in similar materials, the strong hybridization with the p states of the chalcogen atom could play a fundamental role in the proposed valleytronic effects. In this respect, authors should*

show the atom-projected density of states and discuss what is the role of the d-p hybridization, and if it is significant. I would expect that their conclusion would be quite different for systems like WSe₂, where the wider spread of the d-like electronic density of the transition metal would produce a higher hybridization with the p orbitals of the chalcogen atom, the latter thus playing a fundamental role in the proposed valleytronic effects.

Answer: First of all, we completely agree with your opinion that “for systems like WSe₂, the wider spread of the d-like electronic density of the transition metal would produce a higher hybridization with the p orbitals of the chalcogen atom.” However, valleytronics mainly focuses on the electronic behavior near the valleys. Previous research indicated that for classical 2H-TMDs monolayers, the influence of chalcogens’ p-orbitals can be neglected near the valleys **K₊** and **K**. (Ref. [9].: *Phys. Rev. Lett.* **108**, 196802 (2012).). The fact is clearly reflected in the two-band **k·p** model we adopted. As mentioned in our manuscript, “The basis functions are chosen as $|\psi_u^r\rangle = |d_{z^2}\rangle$ and $|\psi_l^r\rangle = (|d_{x^2-y^2}\rangle + i\tau|d_{xy}\rangle) / \sqrt{2}$ ($\tau = \pm 1$ denotes the valley index). The p-orbitals on the chalcogen are neglected in the model.”.

In addition, we have to emphasize that the appearance of **spontaneous valley polarization**, and then **anomalous valley Hall effect** in 2H-TMDs monolayers roots in the coexistence of the **spin-orbit coupling** and the **intrinsic exchange interaction**. The hybridization between the d states of transition-metal and the p states of the chalcogen atoms may affect the dispersion of band structures away from valleys. It would never decide whether the material is of spontaneous valley polarization. In consideration of the nonmagnetic character for classical 2H-TMDs monolayers, like MoS₂, WSe₂, they are certainly not the ferrovalley materials. The anomalous valley Hall effect is naturally absent in such systems.

Figure. R4. The atom-projected band structures of (a) WSe_2 , (b) VSe_2 without ferromagnetism and (c) VSe_2 . Left panels correspond to the occupations of d -orbitals for cation (W or V). Right panels relate to the occupations of p -orbitals for Se atoms. The radius of dots is proportional to its population in certain state. The Fermi level E_F is set to zero in each cases.

Since you are quite interested in the hybridization, we are pleased to compare the atom-projected band structures among WSe_2 , VSe_2 without ferromagnetism and VSe_2 in real

case here. No matter for the WSe₂ (Fig. R4(a)) and the VSe₂ without ferromagnetism (Fig. R4(b)), the bottom of the conduction bands in two valleys dominantly consist from d_{z^2} orbitals on transition metals. At the top of the valance bands, there exist mainly $d_{x^2-y^2}$ and d_{xy} states of cations. The occupations of Se- p states are far beyond the valleys \mathbf{K}_+ and \mathbf{K}_- . Not surprisingly, the energetically degenerated bands between valleys prove that **WSe₂ is merely a paravalley material**. Because the exchange interaction, or equivalently magnetism, is not taken into account, the ferrovalley character for VSe₂ does not appear.

For VSe₂ in real case (Fig. R4(c)), the coexistence between the SOC effect and ferromagnetism splits valley degeneracy, stabilizing the system in ferrovalley state. Although, the dispersion of band structures is quite different from the one excluding magnetism, the hybridization between V- d and Se- p states around the valleys is still negligible.

The atom-projected band structures clearly demonstrate that the electronic behavior near the valleys can be described by the basis functions $|\psi_u^\tau\rangle = |d_{z^2}\rangle$ and $|\psi_l^\tau\rangle = (|d_{x^2-y^2}\rangle + i\tau|d_{xy}\rangle) / \sqrt{2}$ ($\tau = \pm 1$ denotes the valley index). Their hybridization with the p orbitals of the chalcogen atom can be excluded. More importantly, they imply that the coexistence of the SOC effect and exchange interaction of localized d -electrons is the sufficient condition for valley polarization.

Considering that the hybridization is not the key issue of our work, we do not insert the Fig. R4 into our manuscript. We think the sentence “When we ignore the magnetism in monolayer 2H-VSe₂, as shown in Fig. 2a, the band structure is essentially similar to the representative one for TMDs (Fig. 1b).” is strong enough to describe the case. However, if you insist that the figure is necessary, we are also glad to add it.

6) *As a personal suggestion, when authors opt for the "double blind" peer review option, they should remove or modify any sentence in the manuscript that could unveil authors' identity, as suggested in Nature authors' guidelines. For example, in the method section, they say "For the optical property calculations, we adopt our own code OPTICPACK..." providing the reference. Furthermore, if authors want to preserve their anonymity during the peer review process, they should not upload the full version in the arXiv server (see <https://arxiv.org/abs/1604.05833>).*

Answer: The intention to choose the “double blind” peer review option is just for our “curiosity”. Frankly speaking, Nature Communications is the first journal we are aware of providing such option. So, we are willing to be “the first person to try tomato”. As a green hand, we appear to be clumsy and accidentally unveil our identity. However, scientific researches are always continuous. When we introduce the “Method” we adopted during our first-principles calculations, it seems to be inevitable to describe our own code OPTICPACK and then cite corresponding references. Anyway, we apologize for our negligence.

As you can understand, valleytronics is now one of the hottest and most competitive field among condensed matter physics. We are confident that our work is of both fundamentally physical and practically technological importance in this field, as well as some other areas, such as multiferroicity and Hall effects. In order to preserve our credit and broadcast our interesting findings to advocate further research on ferrovalley materials, we upload the full version in the arXiv server in advance. As mentioned above, several experimental groups contacted with us and expressed their interests. The data (http://adsabs.harvard.edu/cgi-bin/nph-ref_history?refs=AR&bibcode=2016arXiv160405833T) from “Reads History” also shows that a broader audience has already been appealed in a very short time. Certainly, the influence of excellent journals like Nature Communications is far beyond it. Therefore, we hope it can be published in Nature Communications as soon as possible.

Following your suggestions, we have tried our best to seriously revise the current manuscript. Hope our earnest responses, listed above, appropriately answer all your concerns, especially the experimental ones, and make you willing to change the decision.

Summary of Changes

1. The new papers have been cited as Ref. Ref.[10].: Kuc, A. & Heine T. The electronic structure calculations of two-dimensional transition-metal dichalcogenides in the presence of external electric and magnetic fields. *Chem. Soc. Rev.* **44**, 2603-2614 (2015); Ref.[36].: Priyanka, M. & Ralph S. 2D transition-metal diselenides: phase segregation, electronic structure, and magnetism. *J. Phys.: Condens. Matter* **28**, 064002 (2016); Ref.[37].: Ma, Y. *et al.* Evidence of the existence of magnetism in pristine VX₂ monolayers (X = S, Se) and their strain-induced tunable magnetic properties. *ACS Nano* **6**, 1695-1701 (2012); Ref.[38].: Atkins, R. *et al.* Synthesis, structure and electrical properties of a new tin vanadium selenide. *J. Solid State Chem.* **202**, 128-133 (2013); Ref.[39].: Yang, J. *et al.* Thickness dependence of the charge-density-wave transition temperature in VSe₂. *Appl. Phys. Lett.* **105**, 063109 (2014); and Ref.[40].: Hite, O. K., Nellist M., Ditto J., Falmbigl M. & Johnson D. C. Transport properties of VSe₂ monolayers separated by bilayers of BiSe. *J. Mater. Res.* **31**, 886-892 (2016).
2. The origin subsection “Anomalous valley Hall effect” now becomes “Discussion”. The concluding paragraph is removed. The sentence “As a new ferroic-family member, its potential coupling with ferroelectric, ferromagnetic, ferroelastic and ferrotoroidic properties may provide novel physics in multiferroic field and promote technological innovation.” has been moved to the fourth paragraph of the main text. The sentence “We strongly advocate experimental efforts on monolayer 2H-VSe₂ and other 2H-phase V-group dichalcogenides, where a series of ferrovalley materials are very likely to hide. It is of great importance in paving the way to the practical applications of valleytronics.” has been transferred to the fourth paragraph of the section “Discussion”.
3. The sentence “With the celebrated discovery of graphene¹, the concept of valleytronics based on graphene-related materials (GRMs) with honeycomb lattice symmetry has attracted immense attention²⁻⁵.” has been corrected as “With the

discovery of graphene¹, the concept of valleytronics has attracted immense attention^{2,3}.”.

4. The sentence “Similar to charge and spin of electrons in electronics and spintronics, the valley degree of freedom in the field of valleytronics, corresponding to degenerate but inequivalent \mathbf{K}_+ and \mathbf{K}_- points (so called valleys) at the corners of the two-dimensional (2D) hexagonal Brillouin zone, constitutes the binary states.” has been simplified as “Similar to charge and spin of electrons in electronics and spintronics, the valley degree of freedom in the field of valleytronics constitutes the binary states.”.
5. “GRMs” and “star of outlook” in the first sentence of the second paragraph have been changed to “valleytronic materials” and “most promising ones”.
6. The sentence “At variance with graphene, the space inversion symmetry for these 2D materials are explicitly broken, which gives rise to the existence of the valley Hall effect⁴, as well as the valley-dependent optical selection rules¹¹.” has been replaced by “With respect to centrosymmetric graphene, the space inversion symmetry for these 2H-phase TMDs is explicitly broken, which gives rise to the existence of the valley Hall effect⁴, as well as the valley-dependent optical selection rules¹¹.”.
7. The sentence “Particularly, the noncentrosymmetry together with intrinsic spin-orbit coupling (SOC) originated from the *d*-orbitals of heavy transition metals¹¹ induce strong coupled spin and valley degree of freedom, making them as a promising platform for the study of the fundamental physics in spintronics, valleytronics and crossing areas.” has been divided into “Particularly, the noncentrosymmetry together with intrinsic spin-orbit coupling (SOC) derived from the *d*-orbitals of heavy transition metals¹² induce strong coupled spin and valley degree of freedom. Therefore, 2H-TMDs monolayers are generally regarded as the promising platform for studies of the fundamental physics in spintronics, valleytronics and crossing areas.”.

8. The sentence “In analogy with paraelectric and paramagnetic materials, the pristine TMDs monolayers are not suitable for long-term storing information.” Has been rewritten as “The pristine TMDs monolayers, however, are not suitable for direct information storage, as the valleys in these systems are not polarized. In analogy to paraelectric and paramagnetic materials, they can be called *paravalley* materials.”.
9. The word “unaccessible” in the sentence “Unfortunately, the extreme field strength for a sizable valley splitting is unaccessible in practical use.” has been corrected as “not accessible”.
10. The grammar mistake in the sentence “When the applied external fields including force, electric, magnetic and optical ones remove, the valleys locked by time-reversal symmetry are still degenerate, stabilizing the system in the initial paravalley state.” Has been corrected. It now becomes “When the applied external fields including force, electric, magnetic and optical ones removed, the valleys locked by time-reversal symmetry are still degenerate, stabilizing the system in the initial paravalley state.”.
11. The sentence “It is known that for representative monolayers of 2H-phase TMDs with trigonal prismatic coordination (D_{3h})^{30,31}, such as MoS₂, the direct band gaps are located at valleys \mathbf{K}_+ and \mathbf{K}_- with C_{3h} point group symmetry.” has been replaced by “For representative monolayers of 2H-phase TMDs, such as MoS₂, they are in trigonal prismatic coordination (D_{3h})^{30,31}. The direct band gaps are located at valleys \mathbf{K}_+ and \mathbf{K}_- with C_{3h} point group symmetry.”.
12. The sentence “A two-band $\mathbf{k}\cdot\mathbf{p}$ model neglecting p -orbitals on the chalcogen with $|\psi_u^\tau\rangle = |d_{z^2}\rangle$ and $|\psi_l^\tau\rangle = (|d_{x^2-y^2}\rangle + i\tau|d_{xy}\rangle) / \sqrt{2}$ ($\tau = \pm 1$ denotes the valley index) as basis functions can be used to describe the electronic properties near the Dirac points \mathbf{K}_\pm ^{4,9}.” has been divided into “A two-band $\mathbf{k}\cdot\mathbf{p}$ model can be used to describe the electronic properties near the Dirac points \mathbf{K}_\pm ^{4,9}. Noted that the basis functions are

chosen as $|\psi_u^r\rangle = |d_z\rangle$ and $|\psi_l^r\rangle = (|d_{x^2-y^2}\rangle + i\tau|d_{xy}\rangle) / \sqrt{2}$ ($\tau = \pm 1$ denotes the valley index). The p -orbitals on the chalcogen are neglected in the model.”.

13. The sentence “According to the total Hamiltonian, the band structures near the valleys \mathbf{K}_\pm of classical TMDs monolayers are easily deduced and schematically drawn in Fig. 1, in which the Fermi level is located at the gap between UB and LB.” has been changed to “According to the total Hamiltonian, the band structures near the valleys \mathbf{K}_\pm of classical TMDs monolayers are easily deduced. They are schematically drawn in Fig. 1, in which the Fermi level is located at the gap between UB and LB.”.
14. The sentence “While for \mathbf{K}_\pm , the one for the bottom of the UB changes as $1E'$.” has been demonstrated much clearer as “While for \mathbf{K}_\pm , they become A' and $1E'$. The IRs for the bottom of UB are different between the two valleys.”.
15. The word “split” in the sentence “It is interesting to point out that E_g^{opt} excited by the left-handed radiation ($E_g^{\text{opt}}(A_+) = \Delta - \lambda_l + \lambda_u + m_l - m_u$) and the one corresponding to the right-handed light ($E_g^{\text{opt}}(B_-) = \Delta - \lambda_l + \lambda_u - m_l + m_u$) split by the magnitude of $2|m_l - m_u|$.” now has been corrected as “are split”.
16. The sentence “Amazingly, inversed chirality of the incident light sees different E_g^{opt} in the valley polarized system, indicating the possibility to judge the valley polarization utilizing noncontact and nondestructive circularly polarized optical means.” has been replaced by “Amazingly, reversed chirality of the incident light sees different E_g^{opt} in the valley polarized system, which indicates the possibility to judge the valley polarization utilizing noncontact and nondestructive circularly polarized optical means.”.
17. The sentence “Above discussions establish the general rule to hunt for ferrovalley materials with spontaneous valley polarization, that is the coexistence of the SOC effect with the intrinsic exchange interaction.” has been slightly modified as “Above results establish the general rule to hunt for ferrovalley materials with spontaneous

valley polarization, i.e. the coexistence of the SOC effect with the intrinsic exchange interaction.”.

18. The phrase “following the tactic” in the sentence “Here, following the tactic, we predict a certain material: 2H-VSe₂ monolayer.” has been replaced by “following the strategy”.
19. The sentence “As a peculiar ferromagnetic semiconductor among TMDs, it possesses intrinsic magnetic moment with the magnitude of $1.01 \mu_B$ in the V-3*d* orbitals, implying remarkable exchange interaction, and then significant spontaneous valley polarization.” has been divided into “As a peculiar ferromagnetic semiconductor among TMDs, it possesses intrinsic magnetic moment with the magnitude of $1.01 \mu_B$ in the V-3*d* orbitals. The strong magnetic coupling implies remarkable exchange interaction, and then significant spontaneous valley polarization.”.
20. The sentence “More excitingly, on the basis of mean field theory and Heisenberg model, its estimated Curie temperature reaches up to ~ 590 K, in accordance with Pan’s work³⁴, declaring that it could be used in valleytronics well above room temperature.” has been divided into “More excitingly, on the basis of mean field theory and Heisenberg model, its estimated Curie temperature reaches up to ~ 590 K, in accordance with Pan’s work³⁵. It demonstrates that 2H-VSe₂ monolayer could be used in valleytronics well above room temperature.”.
21. The new sentence “Noted that the pristine 1T-phase VSe₂ has been widely studied³⁷⁻⁴⁰. However, with the presence of space inversion symmetry, it is not a ferrovalley material.” has been added to the first paragraph of the subsection “Chirality-dependent optical band gap and Berry curvatures in the ferrovalley material”.
22. The sentence “When we ignore the magnetism in monolayer VSe₂, as shown in Fig. 2a, the band structure is essentially similar to the representative one for TMDs (Fig. 1b), except that it is a metal with the Fermi level passing through the states

predominantly comprised of $d_{x^2-y^2}$ and d_{xy} orbitals on cation-V.” has been split to “When we ignore the magnetism in monolayer VSe₂, as shown in Fig. 2a, the band structure is essentially similar to the representative one for TMDs (Fig. 1b). Yet, it is a metal with the Fermi level passing through the states predominantly comprised of $d_{x^2-y^2}$ and d_{xy} orbitals on cation-V.”.

23. The word “Thankfully” in the sentence “Thankfully, the intrinsic exchange interaction of unpaired d electrons, equivalent to a tremendous magnetic field $\sim 1.59 \times 10^4$ T...” has been changed as “Fortunately”.

24. The word “Reversely” in the sentence “Reversely, that of primarily d_{z^2} states is with a relatively greater value at the point \mathbf{K}_+ ($|2m_u - 2\lambda_u| \sim 1.12$ eV) than at \mathbf{K} . ($|2m_u + 2\lambda_u| \sim 1.10$ eV), due to the opposite sign between λ_u and m_u .” has been replaced by “Conversely”.

25. The sentence “When the magnetic moment is inverted, as clearly displayed in Fig. 2d, our interested valley polarization possess reversed polarity, which causes the red shift of E_g^{opt} excited by right-handed light, in comparison to the left-handed one (Fig. 3b).” has been divided into “When the magnetic moment is inverted, as clearly displayed in Fig. 2d, our interested valley polarization possess reversed polarity. As a result, in comparison to the left-handed one, the red shift of E_g^{opt} excited by right-handed light happens (Fig. 3b).”.

26. The word “inversed” in the sentence “Not surprisingly, same value with opposite sign is obtained when the valley polarization has been inversed (Fig. 4d).” has been changed as “reversed”.

27. The sentence “Due to the coexistence of spin and valley Hall current, long-lived spin and valley accumulations on sample sides brings charming phenomena, such as emission of photons with opposite circular polarizations on the two boundaries, and provides a route toward the integration of spintronics and valleytronics⁹.” now has

been rewritten as “Long-lived spin and valley accumulations on sample sides bring charming phenomena, such as emission of photons with opposite circular polarizations on the two boundaries. Moreover, the coexistence of spin and valley Hall current provides a route toward the integration of spintronics and valleytronics⁹”.

28. The word “deriving” in the sentence “We point out that the valley Hall effect in ferrovalley materials possesses a more interesting feature, i.e. the presence of additional charge Hall current deriving from the spontaneous valley polarization.” has been corrected to “originating”.
29. The long sentence “When the polarity of valley reversed, spin-up holes from **K** valley, as net carriers, accumulate in the right side of the sample due to the negative Berry curvature and then lead to measurable transverse voltage with opposite sign.” has been divided into “When the polarity of valley reversed, spin-up holes from **K** valley, as net carriers, accumulate in the right side of the sample due to the negative Berry curvature. Obviously, they lead to measurable transverse voltage with opposite sign.”.
30. A new sentence “The combination of valley, spin and charge accumulations implies the correlation among charge, spin and valley degree of freedom, which makes the new member of Hall family absolutely different from any other forms and attractive in electronics, spintronics, valleytronics and even their crossing areas.” has been added to the penultimate paragraph of the main text.
31. The choice of the PBE functional has been justified as “The exchange-correlation potential is treated in Perdew-Burke-Ernzerhof (PBE) form⁴⁸ of the generalized gradient approximation (GGA) with a kinetic-energy cutoff of 600 eV, as others did³⁴⁻³⁶. We also check that our results are qualitatively robust within the Ceperly-Alder functional form of the local density approximation (LDA) and taking the Hubbard U into account to describe the on-site Coulomb repulsion between $V-d$ electrons.”. The k-mesh used to sample the Brillouin zone now has been stated in our

revised manuscript as “A $18 \times 18 \times 1$ and $36 \times 36 \times 1$ Monkhorst-Pack k -point mesh centered at Γ are respectively adopted in the geometry optimization and self-consistent calculations.”.

32. The meaning of the variables W_k , ν and c in equation 6, which was not demonstrated in the previous version, has been clearly explained as “The integral over the k space has been replaced by a summation over special k points with corresponding weighting factor W_k . The second summation includes ν and c states, based on the reasonable assumption that the VB is fully occupied, while the CB is empty.”.

Some minor changes are not listed here.

Response to Reviewers' Comments (NCOMMS-16-09754A-Z)

Reply to the Reviewer #1

I am very happy with the revision of the manuscript. I think my questions, partially due to fast reading, have improved the presentation a bit.

I would just like the authors to add a little discussion about their material. They write now that two layered phases (T and H) should exist, but a little more information to the reader who is unfamiliar with VSe₂ is needed. The discussion should include (i) stability, in particular inbetween the phases (see e.g. the reference to Zhongfang Chen's work), (ii) potential availability - I found some references to experimental VSe₂, though not to 2H VSe₂ (possibly due to lack of resources as I am travelling). Thus, after this point is addressed (minor revision) I recommend acceptance.

Answer: We rejoice at the positive recommendation from the reviewer as “*after this point is addressed (minor revision) I recommend acceptance*”. And thank him/her very much for the helpful advice, which definitely improves the quality of our manuscript.

Following his/her suggestions, additional discussion about our material 2H-VSe₂ monolayer has been adopted in our revised manuscript. The stability of monolayer VSe₂ between 1T- and 2H-phase has been definitely pointed out as “In addition, Chen’s work³⁴ proved that compared to 1T monolayer, the ferrovalley 2H one is the slightly more stable phase for single-layered VSe₂.” in the end of the first paragraph for the subsection “Chirality-dependent optical band gap and Berry curvatures in the ferrovalley material”. Especially, Zhongfang Chen’s work has been cited here as the reference 34.

When it comes to the potential availability, the 2H-VSe₂ monolayer components were experimentally found in the R polytype VSe₂ bulk. Spiecker et al. (Phys. Rev. Lett. **96**, 086401 (2006).) observed T to R polytype transformation in the thin VSe₂ surface layer, which was induced by Cu deposition. Such transformations were also found when electrochemically intercalating Cu into bulk VSe₂ (K. Sollmann. Ph.D. Thesis, Technical University of Berlin, 1995) or intercalating K into VSe₂ in vacuum (Surf. Sci. **461**, 137 (2000)). In each single-layer component of the R polytype, the V atoms are in trigonal prismatic coordination with Se atoms, representing a 2H-type monolayer. Based on the reviewer's advice, the sentence "components of which have been already found in the R polytype VSe₂ bulk³⁷" has been added to our revised version. The experimental work (Phys. Rev. Lett. **96**, 086401 (2006).) has been cited here as the reference 37.

We are confident that concerns from the reviewer have been well addressed now. By the way, we would like to take this opportunity to thank him/her for the efforts on improving the quality of our work.

Reply to the Reviewer #3

I went through the revised version of the manuscript and I believe that authors made their best effort to answer the reviewers' questions. The work is improved and properly justified. However, a direct experimental application of their formulation is lacking. Despite authors provided examples in their response, those are not quantitative and not directly tied with the quantities present in their formulation. For this reason I cannot still recommend the publication of their work in Nature Communications. Nevertheless, the power of their method and the immediate interest of other experimental groups demonstrate the importance of authors' work. As a such, I renew my suggestion to submit their work to other journals like Physical Review B, or similar; if authors are able to prove more extensively the general result of their

work, I also suggest to submit their manuscript to journal of wider audience like Scientific Report.

Answer: We appreciate that the reviewer acknowledged “*the authors made their best effort to answer the reviewers’ questions. The work is improved and properly justified*”. More importantly, the comments “*the power of their method and the immediate interest of other experimental groups demonstrate the importance of authors’ work*” certainly confirm the novelty and impact of our work. Based on the above and other reviews’ recommendation, we believe that our study, which is of both fundamentally physical and practically technological importance in spintronics, valleytronics and crossing fields, is suitable for publication in Nature Communications, and will appeal to a broader audience.